# SLOT MACHINES: DISCOVERING WINNING COMBINATIONS OF RANDOM WEIGHTS IN NEURAL NETWORKS

## ABSTRACT

In contrast to traditional weight optimization in a continuous space, we demonstrate the existence of effective random networks whose weights are never updated. By selecting a weight among a fixed set of random values for each individual connection, our method uncovers combinations of random weights that match the performance of traditionally-trained networks of the same capacity. We refer to our networks as "slot machines" where each reel (connection) contains a fixed set of symbols (random values). Our backpropagation algorithm "spins" the reels to seek "winning" combinations, i.e., selections of random weight values that minimize the given loss. Quite surprisingly, we find that allocating just a few random values to each connection (e.g., $8$ values per connection) yields highly competitive combinations despite being dramatically more constrained compared to traditionally learned weights. Moreover, finetuning these combinations often improves performance over the trained baselines. A randomly initialized VGG-19 with $8$ values per connection contains a combination that achieves $90\%$ test accuracy on CIFAR-10. Our method also achieves an impressive performance of $98.1\%$ on MNIST for neural networks containing only random weights.

## 1 INTRODUCTION

Innovations in how deep networks are trained have played an important role in the remarkable success deep learning has produced in a variety of application areas, including image recognition (He et al., 2016), object detection (Ren et al., 2015; He et al., 2017), machine translation (Vaswani et al., 2017) and language modeling (Brown et al., 2020). Learning typically involves either optimizing a network from scratch (Krizhevsky et al., 2012), finetuning a pre-trained model (Yosinski et al., 2014) or jointly optimizing the architecture and weights (Zoph & Le, 2017). Against this predominant background, we pose the following question: can a network instantiated with only random weights achieve competitive results compared to the same model using optimized weights?

For a given task, an untrained, randomly initialized network is unlikely to produce good performance. However, we demonstrate that given sufficient random weight options for each connection, there exist selections of these random weight values that have generalization performance comparable to that of a traditionally-trained network with the same architecture. More importantly, we introduce a method that can find these high-performing randomly weighted configurations *consistently* and *efficiently*. Furthermore, we show empirically that a small number of random weight options (e.g., $2-8$ values per connection) are sufficient to obtain accuracy comparable to that of the traditionally-trained network. Instead of updating the weights, the algorithm simply selects for each connection a weight value from a fixed set of random weights.

We use the analogy of "slot machines" to describe how our method operates. Each reel in a Slot Machine has a fixed set of symbols. The reels are jointly spinned in an attempt to find winning combinations. In our context, each connection has a fixed set of random weight values. Our algorithm "spins the reels" in order to find a winning combination of symbols, i.e., selects a weight value for each connection so as to produce an instantiation of the network that yields strong performance. While in physical Slot Machines the spinning of the reels is governed by a fully random process, in our Slot Machines the selection of the weights is guided by a method that optimizes the given loss at each spinning iteration.

More formally, we allocate $K$ fixed random weight values to each connection. Our algorithm assigns a quality score to each of these $K$ possible values. In the forward pass a weight value is selected for each connection based on the scores. The scores are then updated in the backward pass via stochastic gradient descent. However, the weights are never changed. By evaluating different combinations of fixed randomly generated values, this extremely simple procedure finds weight configurations that yield high accuracy.

We demonstrate the efficacy of our algorithm through experiments on MNIST and CIFAR-10. On MNIST, our randomly weighted Lenet-300-100 (Lecun et al., 1998) obtains a $97.0\%$ test set accuracy when using $K = 2$ options per connection and $98.1\%$ with $K = 128$. On CIFAR-10 (Krizhevsky, 2009), our six-layer convolutional network matches the test set performance of the traditionally-trained network when selecting from $K = 64$ fixed random values at each connection.

Finetuning the models obtained by our procedure generally boosts performance over networks with optimized weights albeit at an additional compute cost (see Figure 4). Also, compared to traditional networks, our networks are less memory efficient due to the inclusion of scores. That said, our work casts light on some intriguing phenomena about neural networks:

- First, our results suggest a performance comparability between selection from multiple random weights and traditional training by continuous weight optimization. This underscores the effectiveness of strong initializations.

- Second, this paper further highlights the enormous expressive capacity of neural networks. Maennel et al. (2020) show that contemporary neural networks are so powerful that they can memorize randomly generated labels. This work builds on that revelation and demonstrates that current networks can model challenging non-linear mappings extremely well even by simple selection from random weights.

- This work also connects to recent observations (Malach et al., 2020; Frankle & Carbin, 2018) suggesting strong performance can be obtained by utilizing gradient descent to uncover effective subnetworks.

- Finally, we are hopeful that our novel model —consisting in the the introduction of multiple weight options for each edge— will inspire other initialization and optimization strategies.

## 2 RELATED WORK

**Supermasks and the Strong Lottery Ticket Conjecture.** The lottery ticket hypothesis was articulated in (Frankle & Carbin, 2018) and states that a randomly initialized neural network contains sparse subnetworks which when trained in isolation from scratch can achieve accuracy similar to that of the trained dense network. Inspired by this result, Zhou et al. (2019) present a method for identifying subnetworks of randomly initialized neural networks that achieve better than chance performance without training. These subnetworks (named "supermasks") are found by assigning a probability value to each connection. These probabilities are used to sample the connections to use and are updated via stochastic gradient descent. Without ever modifying the weights, Zhou et al. (2019) find subnetworks that perform impressively across multiple datasets.

Follow up work by Ramanujan et al. (2019) finds supermasks that match the performance of a dense network. On ImageNet (Russakovsky et al., 2009), they find subnetworks within a randomly weighted ResNet-50 (Zagoruyko & Komodakis, 2016) that match the performance of a smaller, trained ResNet-34 (He et al., 2016). Accordingly, they propose the strong lottery ticket conjecture: a sufficiently overparameterized, randomly weighted neural network contains a subnetwork that performs as well as a traditionally-trained network with the same number of parameters. Ramanujan et al. (2019) adopts a deterministic protocol in their so-called "edge-popup" algorithm for finding supermasks instead of the stochastic algorithm of Zhou et al. (2019).

These empirical results as well as recent theoretical ones (Malach et al., 2020; Pensia et al., 2020) suggest that pruning a randomly initialized network is just as good as optimizing the weights, provided a good pruning mechanism is used. Our work corroborates this intriguing phenomenon but differs from these prior methods in a significant aspect. We eliminate pruning completely and instead introduce multiple weight values per connection. Thus, rather than selecting connections to define a subnetwork, our method selects weights for all connections in a network of fixed structure.

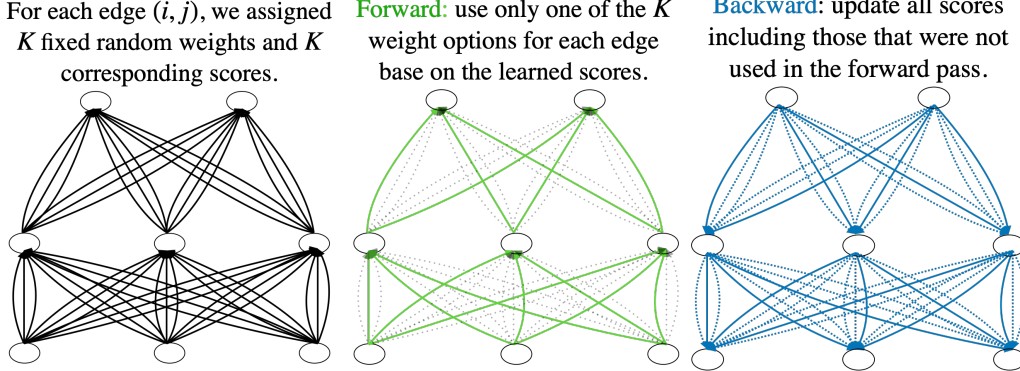

For each edge $(i, j)$, we assigned $K$ fixed random weights and $K$ corresponding scores.

Forward: use only one of the $K$ weight options for each edge base on the learned scores.

Backward: update all scores including those that were not used in the forward pass.

Figure 1: Our method assigns a set of $K$ ($K = 3$ in this illustration) random weight options to each connection. During the forward pass, one of the $K$ values is selected for each connection, based on a quality score computed for each weight value. On the backward pass, the quality scores of all weights are updated using a straight-through gradient estimator (Bengio et al., 2013), enabling the network to sample better weights in future passes. Unlike the scores, the weights are never changed.

Although our work has interesting parallels with pruning (as illustrated in Figure 9), it is different from pruning as all connections remain active in every forward pass.

**Pruning at Initialization.** The lottery ticket hypothesis also inspired several recent work aimed towards pruning (i.e., predicting "winning" tickets) at initialization (Lee et al., 2020; 2019; Tanaka et al., 2020; Wang et al., 2020). Our work is different in motivation from these methods and those that train only a subset of the weights (Hoffer et al., 2018; Rosenfeld & Tsotsos, 2019). Our aim is to find neural networks with random weights that match the performance of traditionally-trained networks with the same number of parameters.

**Weight Agnostic Neural Networks.** Gaier & Ha (2019) build neural network architectures with high performance in a setting where all the weights have the same shared random value. The optimization is instead performed over the architecture (Stanley & Miikkulainen, 2002). They show empirically that the network performance is indifferent to the shared value but defaults to random chance when all the weights assume different random values. Although we do not perform weight training, the weights in this work have different random values. Further, we build our models using fixed architectures.

**Low-bit Networks and Quantization Methods.** As in binary networks (Courbariaux & Bengio, 2016; Rastegari et al., 2016) and network quantization (Hubara et al., 2017; Wang et al., 2018), the parameters in slot machines are drawn from a finite set. However, whereas the primary objective in quantized networks is mostly compression and computational speedup, the motivation behind slot machines is recovering good performance from randomly initialized networks. Accordingly, slot machines use real-valued weights as opposed to the binary (or integer) weights used by low-bit networks. Furthermore, the weights in low-bit networks are usually optimized directly whereas only associated scores are optimized in slot machines.

**Random Decision Trees.** Our approach is inspired by the popular use of random subsets of features in the construction of decision trees (Breiman et al., 1984). Instead of considering all possible choices of features and all possible splitting tests at each node, random decision trees are built by restricting the selection to small random subsets of feature values and splitting hypotheses. We adapt this strategy to the training of neural network by restricting the optimization of each connection over a random subset of weight values.

## 3 SLOT MACHINES: NETWORKS WITH FIXED RANDOM WEIGHT OPTIONS

Our goal is to construct non-sparse neural networks that achieve high accuracy by selecting a value from a fixed set of completely random weights for each connection. We start by providing an intuition for our method in Section 3.1, before formally defining our algorithm in Section 3.2 .

## 3.1 INTUITION

An untrained, randomly initialized network is unlikely to perform better than random chance. Interestingly, the impressive advances of Ramanujan et al. (2019) and Zhou et al. (2019) demonstrate that networks with random weights can in fact do well, if pruned properly. In this work, instead of pruning we explore weight selection from fixed random values as a way to obtain effective networks. To provide an intuition for our method, consider an untrained network $N$ with one weight value for each connection, as typically done. If the weights of $N$ are drawn randomly from an appropriate distribution $\mathcal{D}$ (e.g., (Glorot & Bengio, 2010) or (He et al., 2015)), there is an extremely small but nonzero probability that $N$ obtains good accuracy (say, greater than a threshold $\tau$) on the given task. Let $q$ denote this probability. Also consider another untrained network $N_K$ that has the same architectural configuration as $N$ but with $K > 1$ weight choices per connection. If $n$ is the number of connections in $N$, then $N_K$ contains within it $K^n$ different network instantiations that are architecturally identical to $N$ but that differ in weight configuration. If the weights of $N_K$ are sampled from $\mathcal{D}$, then the probability that none of the $K^n$ networks obtains good accuracy is essentially $(1-q)^{K^n}$. This probability decays quickly as either $K$ or $n$ increases. Our method finds randomly weighted networks that achieve very high accuracy even with small values of $K$. For instance, a six layer convolutional network with 2 random values per connection obtains $82\%$ test accuracy on CIFAR-10.

But how do we select a good network from these $K^n$ different networks? Brute-force evaluation of all possible configurations is clearly not feasible due to the massive number of different hypotheses. Instead, we present an algorithm, shown in Figure 1, that iteratively searches the best combination of connection values for the entire network by optimizing the given loss. To do this, the method learns a real-valued quality score for each weight option. These scores are used to select the weight value of each connection during the forward pass. The scores are then updated in the backward pass based on the loss value in order to improve training performance over iterations.

## 3.2 LEARNING IN SLOT MACHINES

Here we introduce our algorithm for the case of fully-connected networks but the description extends seamlessly to convolutional networks. A fully-connected neural network is an acyclic graph consisting of a stack of $L$ layers $[1, \cdots, L]$ where the $\ell$th layer has $n_\ell$ neurons. The activation $h(x)_i^{(\ell)}$ of neuron $i$ in layer $\ell$ is given by

$$h(x)_i^{(\ell)} = g\left(\sum_{j=1}^{n_{\ell-1}} h(x)_j^{(\ell-1)} W_{ij}^{(\ell)}\right) \tag{1}$$

where $W_{ij}^{(\ell)}$ is the weight of the connection between neuron $i$ in layer $\ell$ and neuron $j$ in layer $\ell - 1$, $x$ represents the input to the network, and $g$ is a non-linear function. Traditionally, $W_{ij}^{(\ell)}$ starts off as a random value drawn from an appropriate distribution before being optimized with respect to a dataset and a loss function using gradient descent. In contrast, our method does not ever update the weights. Instead, it associates a set of $K$ possible weight options for each connection[1], and then it optimizes the selection of weights to use from these predefined sets for all connections.

**Forward Pass.** Let $\{W_{ij1}, \ldots, W_{ijK}\}$[2] be the set of the $K$ possible weight values for connection $(i, j)$ and let $s_{ijk}$ be the "quality score" of value $W_{i,j,k}$, denoting the preference for this value over the other possible $K - 1$ values. We define a selection function $\rho$ which takes as input the scores $\{s_{ij1}, \ldots, s_{ijK}\}$ and returns an index between 1 and $K$. In the forward pass, we set the weight of $(i, j)$ to $W_{ijk^*}$ where $k^* = \rho(s_{ij1}, \ldots, s_{ijK})$.

In our work we set $\rho$ to be either the $\arg\max$ function (returning the index corresponding to the largest score) or the sampling from a Multinomial distribution defined by $\{s_{ij1}, \ldots, s_{ijK}\}$. We refer to the former as Greedy Selection (GS). We name the latter Probabilistic Sampling (PS) and

---

[1]For simplicity, we use the same number of weight options $K$ for all connections in a network.
[2]For brevity, from now on we omit the superscript denoting the layer.

(a) 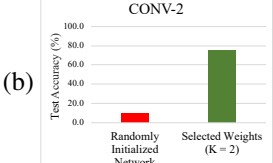 (b) 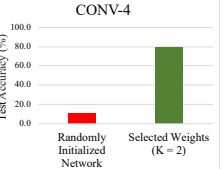 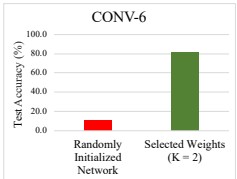

Figure 2: Selecting from only $K = 2$ weight options per connection already dramatically improves accuracy compared to an untrained network that performs at random chance ($10\%$) on both (a) MNIST and (b) CIFAR-10. The first bar in each plot shows the performance of an untrained randomly initialized network and the second bar shows the results of selecting random weights with GS using $K = 2$ options per connection.

implement it as

$$\rho(s_{ij1}, \ldots, s_{ijK}) \sim \text{Mult}\left(\frac{e^{s_{ij1}}}{\sum_{k=1}^{K} e^{s_{ijk}}}, \frac{e^{s_{ij2}}}{\sum_{k=1}^{K} e^{s_{ijk}}}, \cdots, \frac{e^{s_{ijK}}}{\sum_{k=1}^{K} e^{s_{ijk}}}\right). \tag{2}$$

The empirical comparison between these two selection strategies is given in Section 4.5.

We note that, although $K$ values per connection are considered during training (as opposed to the infinite number of possible values of traditional training), only one value per connection is used at test time. The final network is obtained by selecting for each connection the value corresponding to the highest score (for both GS and PS) upon completion of training. Thus, the effective capacity of the network at inference time is the same as that of a traditionally-trained network.

**Backward Pass.** In the backward pass, all the scores are updated with straight-through gradient estimation since $\rho$ has a zero gradient almost everywhere. The straight-through gradient estimator (Bengio et al., 2013) treats $\rho$ essentially as the identity function in the backward pass by setting the gradient of the loss with respect to $s_{ijk}$ as

$$\nabla_{s_{ijk}} \leftarrow \frac{\partial \mathcal{L}}{\partial a(x)_i^{(\ell)}} h(x)_j^{(\ell-1)} W_{ijk}^{(\ell)} \tag{3}$$

for $k \in \{1, \cdots, K\}$ where $\mathcal{L}$ is the objective function. $a(x)_i^{(\ell)}$ is the pre-activation of neuron $i$ in layer $\ell$. Given $\alpha$ as the learning rate, and ignoring momentum, we update the scores via stochastic gradient descent as

$$\tilde{s}_{ijk} = s_{ijk} - \alpha \nabla_{s_{ijk}} \tag{4}$$

where $\tilde{s}_{ijk}$ is the score after the update. Our experiments demonstrate that this simple algorithm learns to select effective configurations of random weights resulting in impressive results across different datasets and models.

## 4 Experiments

### 4.1 Experimental Setup

The weights of all our networks are sampled uniformly at random from a Glorot Uniform distribution (Glorot & Bengio, 2010), $\mathbb{U}(-\sigma_x, \sigma_x)$ where $\sigma_x$ is the standard deviation of the Glorot Normal distribution. We ignore $K$, the number of options per connection, when computing the standard deviation since it does not affect the network capacity. Like for the weights, we initialize the scores independently from a uniform distribution $\mathbb{U}(\gamma, \gamma + \lambda\sigma_x)$ where $\gamma$ and $\lambda$ are small constants. We use $\lambda = 0.1$ for all fully-connected layers and set $\lambda$ to 1 when initializing convolutional layers. We always set $\gamma$ to 0. We use $15\%$ and $10\%$ of the training sets of MNIST and CIFAR-10, respectively, for validation. We report performance on the separate test set. On MNIST, we experiment with the Lenet-300-100 (Lecun et al., 1998) architecture following the protocol in Frankle & Carbin (2018). We also use the VGG-like architectures used thereof and in Zhou et al. (2019) and Ramanujan et al. (2019). We denote these networks as CONV-2, CONV-4, and CONV-6. For completeness, these architectures and their training schedules are provided in Appendix C. All our plots show the averages of five different independent trials. Error bars whenever shown are the minimum and maximum

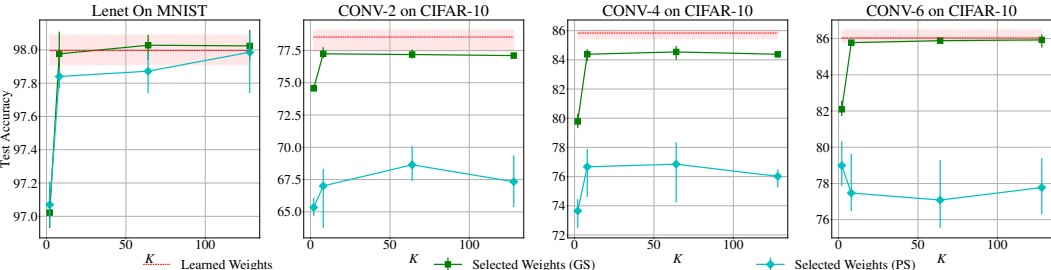

Figure 3: **Comparison with traditional training on CIFAR-10 and MNIST.** Performance of Slot Machines improves by increasing $K$ (here we consider $K \in \{2, 8, 64, 128\}$) although the gains after $K \geq 8$ are small. For CONV-6 (the deepest model considered here), our approach using GS achieves accuracy comparable to that obtained with trained weights, while for CONV-2 and CONV-4 it produces performance only slightly inferior to that of the optimized network. Furthermore, as illustrated by the error bars in these plots, the accuracy variances of Slot Machines with GS are much smaller than those of networks traditionally trained by optimizing weights. Accuracies are measured on the *test* set over five different trials using early stopping on the *validation* accuracy with a horizon of 30 epochs for all models.

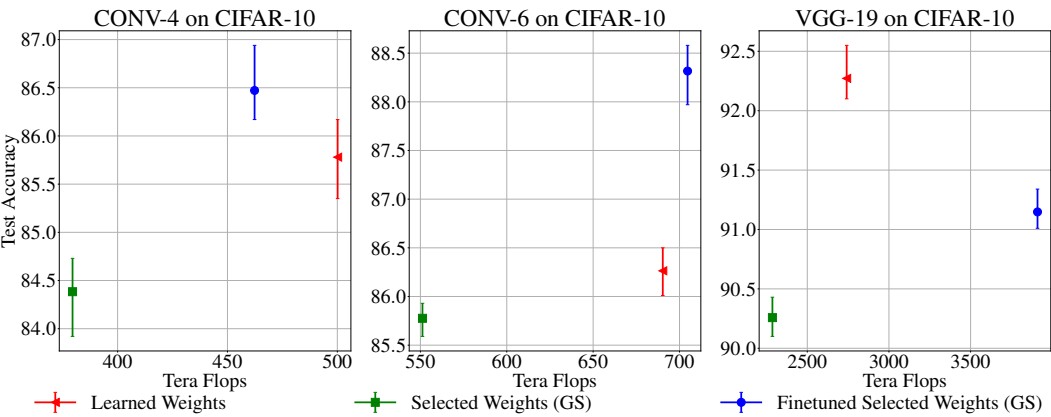

Figure 4: **Finetuning Selected Weights.** Finetuning Slot Machines improves test set performance on CIFAR-10. For CONV-4 and CONV-6 this results in better accuracy compared to the same networks learned from scratch at comparable training cost (shown on the $x$ axis). All Slot Machines here use $K = 8$ options per edge.

over the trials. A core component of our algorithm is the hyperparameter $K$ which represents the number of options per connection. As such, we conduct experiments with $K \in \{2, 8, 64, 128\}$ and analyze the performance as $K$ varies. When not otherwise noted, we used the Greedy Selection (GS) method. We empirically compare GS to Probabilistic Selection (PS) in 4.5.

## 4.2 SLOT MACHINES VERSUS TRADITIONALLY-TRAINED NETWORKS

We compare the networks using random weights selected from our approach with two different baselines: (1) randomly initialized networks with one weight option per connection, and (2) traditionally-trained networks whose weights are optimized directly. These baselines are off-the-shelf modules from PyTorch (Paszke et al., 2019) which we train in the standard way according to the schedules presented in Appendix C.

As shown in Figure 2, untrained dense networks perform at chance, if they have only one weight per edge. However, methodologically selecting the parameters from just two random values for each connection greatly enhances performance across different datasets and networks. Even better, as shown in Figure 3, as the number of random weight options per connection increases, the performance of these networks approaches that of traditionally-trained networks with the same number of

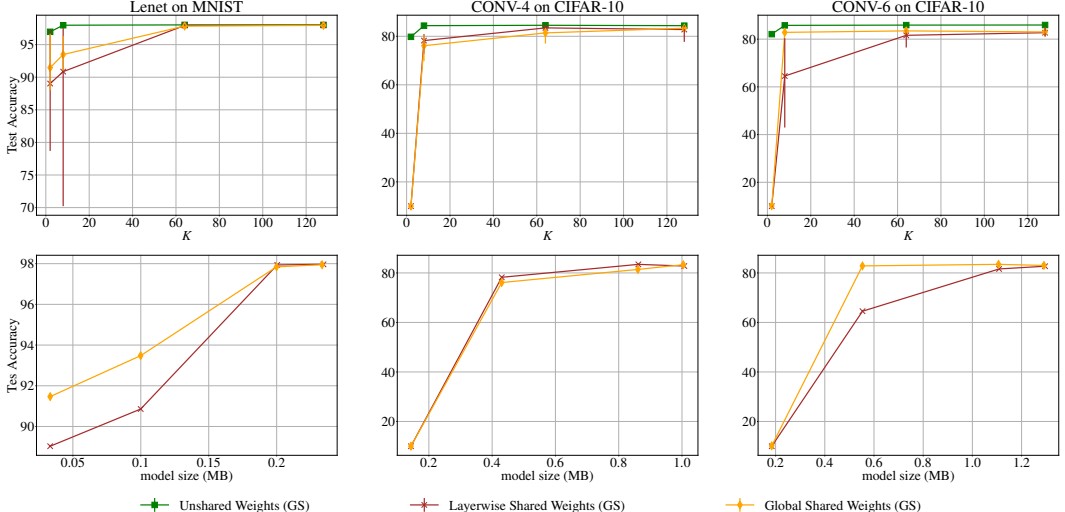

Figure 5: **Sharing random weights**: Slot Machines using the same set of $K$ random weights for all connections in a layer or even in the entire network perform quite well. However, they do not match the performance of Slot Machines that use different sets of weights for different connections. The benefit of sharing weights is that the space needed to store these networks is substantially smaller compared to the storage footprint of slot machines with unshared weights[3]. We do not include the curves for slot machines with unshared weights in the second row because they are disproportionately too large relative to the shown plots. As an example, a Lenet model with unshared weights needs $\sim 1$MB of storage.

parameters, despite containing only random values. Malach et al. (2020) proved that any "ReLU network of depth $\ell$ can be approximated by finding a weighted-subnetwork of a random network of depth $2\ell$ and sufficient width." Without pruning, our selection method finds within the superset of fixed random weights a 6 layer configuration that performs as well as a 6 layer traditionally-trained network.

## 4.3 FINETUNING SLOT MACHINES

Our approach can also be viewed as a strategy to provide a better initialization for traditional training. To assess the value of such a scheme, we finetune the networks obtained after training Slot Machines for 100 epochs (see Appendix C for implementation details). Figure 4 summarizes the results in terms of training time (including both selection and finetuning) vs test accuracy. It can be noted that for the CONV-4 and CONV-6 architectures, finetuned Slot Machines achieve higher accuracy compared to the same models learned from scratch, effectively at no additional training cost. For VGG-19, finetuning improves accuracy, but the resulting model still does not match the performance of the model trained from scratch.

To show that the weight selection in Slot Machines does in fact impact performance of finetuned models, we finetune from different Slot Machine checkpoints. If the selection is beneficial, then finetuning from later checkpoints will show improved performance. As shown in Figure 6, this is indeed the case as finetuning from later Slot Machine checkpoints results in higher performance on the test set.

## 4.4 SHARING RANDOM WEIGHTS

Inspired by quantized networks (Courbariaux & Bengio, 2016; Rastegari et al., 2016; Hubara et al., 2017; Wang et al., 2018), in this experiment we consider slot machines under two different settings. The first constrains the connections in a layer to share the same set of $K$ random weights. The second

---

[3]The set of shared values for the weights can be stored in a lookup table and thus the selected weight for each connection is compactly represented by a low-bit index into the table.

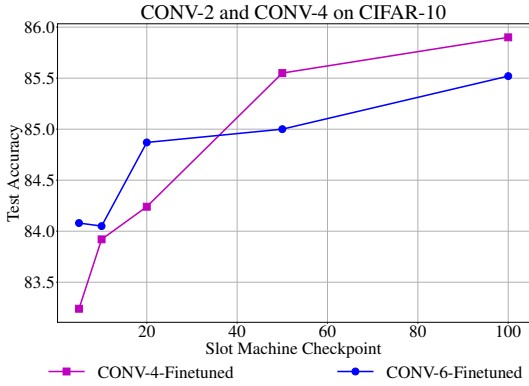

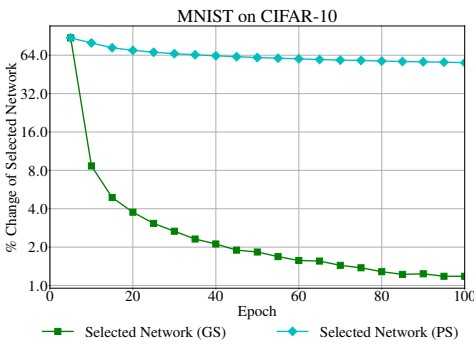

Figure 6: **Finetuning from different Slot Machine checkpoints.** Slot Machine checkpoint shows the number of training epochs used for weight selection before switching to finetuning (performed for 100 epochs). Performance is measured on the test set using early stopping determined by the maximum validation accuracy during finetuning.

Figure 7: **Weight exploration in Slot Machines.** The vertical axis shows (on a log scale) the percentage of weights changed after every five epochs as training progresses. It can be seen that, compared to PS, GS is much less exploratory and converges rapidly to a preferred configuration of weights. On the other hand, due to its probabilistic sampling, PS keeps changing the weight selections even in late stages of training.

setting is even more restricting by requiring all connections in the network to share the same set of $K$ random weights. Under the first setting, at each layer the weights are drawn from the uniform distribution $\mathbb{U}(-\sigma_\ell, \sigma_\ell)$ where $\sigma_\ell$ is the standard deviation of the Glorot Normal distribution for layer $\ell$. When using a single set of weights for the entire network, we sample the weights independently from $\mathbb{U}(-\hat{\sigma}, \hat{\sigma})$. $\hat{\sigma}$ is the mean of the standard deviations of the per layer Glorot Normal distributions.

Each of the weights is still associated with a score. The slot machine with shared weights is then trained as before. This approach has the potential of compressing the model although the full set of of scores is still needed.

As shown in Figure 5, these models continue to do well when $K$ is large enough. However, unlike conventional slot machines, these models do not do well when $K$ is very small, e.g., $K = 2$. Furthermore, when $K \leq 64$, the accuracy exhibits a large variance from run to run, as evidenced by the large error bars in the plot. This is understandable, as the slot machine with shared weights is restricted to search in a much smaller space of parameter combinations and thus the probability of finding a winning combination is greatly reduced.

## 4.5 GREEDY SELECTION VERSUS PROBABILISTIC SAMPLING

As detailed in Section 3.2, we consider two different methods for sampling our networks in the forward pass: a greedy selection where the weight corresponding to the highest score is used and a stochastic selection which draws from a proper distribution over the weights. We compare the behavior of our networks under these two different protocols.

As seen in Figure 3, GS performs better. To fully comprehend the performance differences between these two strategies, it is instructive to look at Figure 7, which reports the percentage of weights changed every 5 epochs by the two strategies. PS keeps changing a large percentage of weights even in late stages of the optimization, due to its probabilistic sampling. Despite the network changing considerably, PS still manages to obtain good accuracy (see Figure 3) indicating that there are potentially many good random networks within a Slot Machine. However, as hypothesized in Ramanujan et al. (2019), the high variability due to stochastic sampling means that the same network is likely never or rarely observed more than once in any training run. This makes learning extremely challenging and consequently adversely impacts performance. Conversely, GS is less exploratory and converges fairly quickly to a stable set of weights.

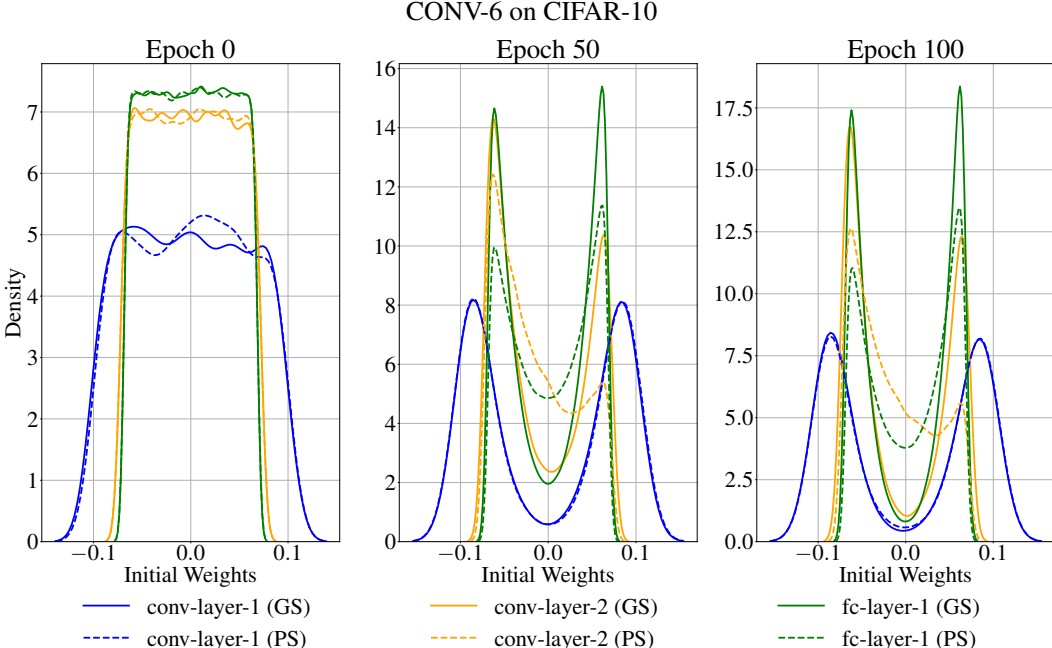

Figure 8: The distributions of the selected weights in the first two convolutional and the first fully-connected layers of CONV-6 on CIFAR-10. Starting from purely uniform distributions, Slot Machines progressively choose large magnitude weights as training proceeds. See Appendix D for plots showing similar behavior in other Slot Machines.

From Figure 3, we can also notice that the accuracy of GS improves or remains stable as the value of $K$ is increased. This is not always the case for PS when $K \geq 8$. We claim this behavior is expected since GS is more restricted in terms of the choices it can take. Thus, GS benefits more from large values of $K$ compared to PS.

## 4.6 DISTRIBUTION OF SELECTED WEIGHTS

Here we look at the distribution of selected weights at different training points in order to understand why certain weights are chosen and others are not. As shown in Figure 8, both GS and PS tend to prefer weights having large magnitudes as learning progresses. This propensity for large weights might help explain why methods such as magnitude-based pruning of traditionally-trained networks work as well as they do. Unlike the weights, we find that the scores associated to the selected weights form a normal distribution overtime as shown in Figure 12 in the Appendix.

## 5 CONCLUSION AND FUTURE WORK

This work shows that neural networks with random weights perform competitively, provided that each connection is given multiple weight options and that a good selection strategy is used. We introduce a simple selection procedure that is remarkably effective and consistent in producing strong weight configurations from few random options per connection. We also demonstrate that these selected configurations can be used as starting initializations for finetuning, which often produces accuracy gains over training the network from scratch, at comparable computational cost. Our study suggests that our method tends to naturally select large magnitude weights as training proceeds. Future work will be devoted to further analyze what other properties differentiate selected weights from those that are not selected, as knowing such properties may pave the way for more effective initializations for neural networks.

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

## A    RELATION TO PRUNING

The models learned by our algorithm could in principle be found by applying pruning to a bigger network representing the multiple weight options in the form of additional connections. One way to achieve this is by introducing additional "dummy" layers after every layer $\ell$ except the output layer. Each "dummy" layer will have $K * c$ identity units where $c = n_\ell * n_{\ell+1}$ and $n_\ell$ is the number of neurons in layer $\ell$. The addition of these layers has the effect of separating out the $K$ random values for each connection in our network into distinct connection weights. It is important that the neurons of the "dummy" layer encode the identity function to ensure that the random values can pass through it unmodified. Finally, in order to obtain the model learned by our system, all connections between a layer and its associated "dummy" layer must be pruned except for the weights which would have been selected by our algorithm as shown in Figure 9. This procedure requires allocating a bigger network and is clearly more costly compared to our algorithm.

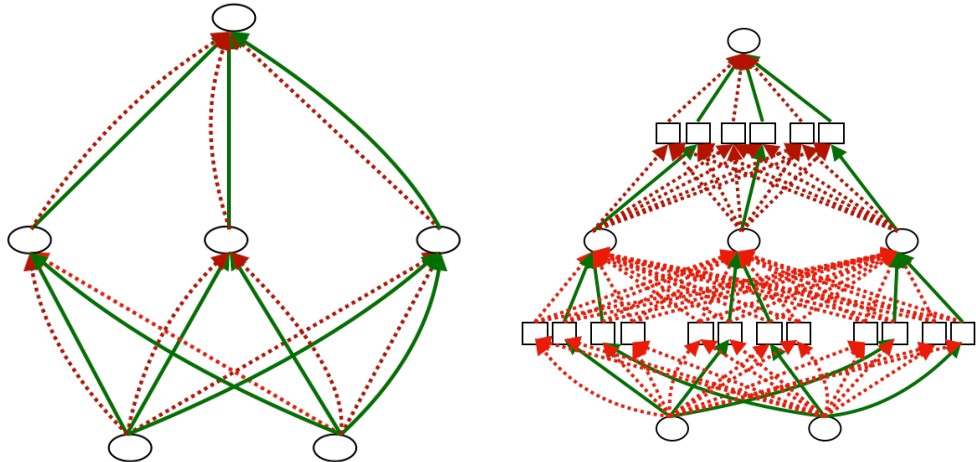

Figure 9: In principle, it is possible to obtain our network (*left*) by pruning a bigger network constructed ad-hoc (*right*). In this example, our slot machine uses $K = 2$ options per connection $(i, j)$. The green-colored connections represent the selected weights. The square boxes in the bigger network implement identity functions. The circles designate vanilla neurons with non-linear activation functions. Red dash lines in our network represent unchosen weights. These lines in the bigger network would correspond to pruned weights.

## B    EXPERIMENTAL COMPARISON WITH PRIOR PRUNING APPROACHES

Our approach is similar to the pruning technique of Ramanujan et al. (2019) as their method too does not update the weights of the networks after initialization. Furthermore, their strategy selects the weights greedily, as in our GS. However, they use one weight per connection and employ pruning to uncover good subnetworks within the random network whereas we use multiple random values per edge. Additionally, we do not ever prune any of the connections. We compare the results of our networks to this prior work in Table 1. We also compare with supermasks (Zhou et al., 2019). Supermasks employ a probability distribution during the selection which makes them reminiscent of our PS models. However, they use a Bernoulli distribution at each weight while PS uses a Multinomial distribution at each connection. Also, like Ramanujan et al. (2019), supermasks have one weight per connection and perform pruning rather than weight selection. Table 1 shows that GS achieves accuracy comparable to that of Ramanujan et al. (2019) while PS matches the performance of supermasks. These results suggest an interesting empirical performance equivalency among these related but distinct approaches.

Table 1: Comparison with Ramanujan et al. (2019) and (Zhou et al., 2019). The results of the first two rows are from the respective papers.

| Method | Lenet | CONV-2 | CONV-4 | CONV-6 |
|---|---|---|---|---|
| Ramanujan et al. (2019) | - | 77.7 | 85.8 | 88.1 |
| SNIP (Zhou et al., 2019) | 98.0 | 66.0 | 72.5 | 76.5 |
| Slot Machines (GS) | 98.1 | 77.2 | 84.6 | 86.3 |
| Slot Machines (PS) | 98.0 | 67.8 | 76.7 | 78.3 |

Table 2: Architecture specifications of the networks used in our experiments. The Lenet network is trained on MNIST. The CONV-$x$ models are the same VGG-like architectures used in (Frankle & Carbin, 2018; Zhou et al., 2019; Ramanujan et al., 2019). All convolutions use $3 \times 3$ filters and pool denotes max pooling.

| *Network* | Lenet | CONV-2 | CONV-4 | CONV-6 | VGG-19 |
|---|---|---|---|---|---|
| *Convolutional Layers* | | $64, 64$, pool | $64, 64$, pool
$128, 128$, pool | $64, 64$, pool
$128, 128$, pool
$256, 256$, pool | 2x64, pool
2x128, pool
2x256, pool
4x512, pool
4x512, avg-pool |
| *Fully-connected Layers* | $300, 100, 10$ | $256, 256, 10$ | $256, 256, 10$ | $256, 256, 10$ | 10 |
| *Epochs:* GS | 140 | 160 | 140 | 140 | 100 |

## C  FURTHER IMPLEMENTATION DETAILS

We find that a high learning rate is required when sampling the network probabilistically—a behaviour which was also observed in Zhou et al. (2019). Accordingly, we use a learning rate of 25 for all PS models except the six layer convolutional network where we use a learning rate of 50. We did not train VGG-19 using a probabilistic selection. Also, a slightly lower learning rate helps GS networks when $K \geq 64$. Accordingly, we set a constant learning rate of 0.1 for GS models for $K \in \{2, 8\}$ and 0.01 for $K \in \{64, 128\}$. When optimizing the weights directly from scratch or fine-tuning, we decay the learning rate by a factor of $10\%$ at epoch 20 for all models except when training CONV-2 and CONV-4 from scratch where we decay the rate at epoch 60. Learned weights and fine-tuned models all use the same initial learning rate of 0.01. All the VGG-19 models use an initial learning rate of 0.1 which is decayed by a factor of $10\%$ at epoch 80 and additionally at epoch 120 when directly optimizing the weights from scratch. Finetuning in VGG-19 also starts with an initial learning rate of 0.1 which is reduced by a factor of 10 at epoch 20.

Whenever we sample using PS, we train for additional 40 epochs relative to the number of epochs used by the corresponding GS model as shown in Table 2. We do this to compensate for the slow convergence of PS models as shown in Figure 7. All finetuned models are trained for 100 epochs. When directly optimizing the weights from scratch, we set the number of epochs to be the sum of the number of epochs used for the corresponding selection checkpoint and that of the finetuned model; we then report the test accuracy based on early stopping with a horizon of 30 with respect to the validation accuracy.

We use data augmentation and dropout (with a rate of $p = 0.5$) when experimenting on CIFAR-10 (Krizhevsky, 2009). All models use a batch size of 128 and stochastic gradient descent optimizer with a momentum of 0.9. We do not use weight decay when optimizing slot machines (for both GS and PS ). But we do use a weigh decay of $1e - 4$ for all directly optimized models (training from scratch and finetuning). We use batch normalization in VGG-19 but the affine parameters are never updated throughout training.

# D DISTRIBUTION OF SELECTED WEIGHTS AND SCORES

As discussed in Section 4.6 and shown in Figure 8, we observe that slot machines tend to choose increasingly large magnitude weights as learning proceeds. In Figures 10, 11, 12 of this appendix, we show similar plots for other models. We reasoned that the observed behavior might be due to the Glorot Uniform distribution from which the weights are sampled. Accordingly, we performed ablations for this where we used a Glorot Normal distribution for the weights as opposed to the Glorot Uniform distribution used throughout the paper. As shown in Figure 10a, the initialization distribution do indeed contribute to observed pattern of preference for large magnitude weights. However, initialization may not be the only reason as the models continue to choose large magnitude weights even when the weights are sampled from a Glorot Normal distribution. This is shown more clearly in the third layer of Lenet which has relatively fewer weights compared to the first two layers. We also observed a similar behavior in normally distributed convolutional layers.

Different from the weights, notice that the selected scores are distributed normally as shown in Figure 12. The scores in PS move much further away from the initial values compared to those in GS. This is largely due to the large learning rates used in PS models.

# E SCORES INITIALIZATION

We initialize the quality scores by sampling from a uniform distribution $\mathbb{U}(\gamma, \gamma + \lambda\sigma_x)$. As shown in Figure 13, we observe that our networks are sensitive to the range of the uniform distribution the scores are drawn from when trained using GS. However, as expected we found them to be insensitive to the position of the distribution $\gamma$. Generally, narrow uniform distributions, e.g., $\mathbb{U}(0, 0.1)$, lead to higher test set accuracy compared to wide distributions e.g., $\mathbb{U}(0, 1)$. This matches intuition since the network requires relatively little effort to drive a very small score across a small range compared to a large range. To concretize this intuition, take for example a weight $\tilde{w}$ that gives the minimum loss for connection $(i, j)$. If its associated score $\tilde{s}$ is initialized poorly to a small value, and the range is small, the network will need little effort to push it to the top to be selected. However, if the range is large, the network will need much more effort to drive $\tilde{s}$ to the top for $\tilde{w}$. We believe that this sensitivity to the distribution range could be compensated by using higher learning rates for wider distributions of scores and vice-versa.

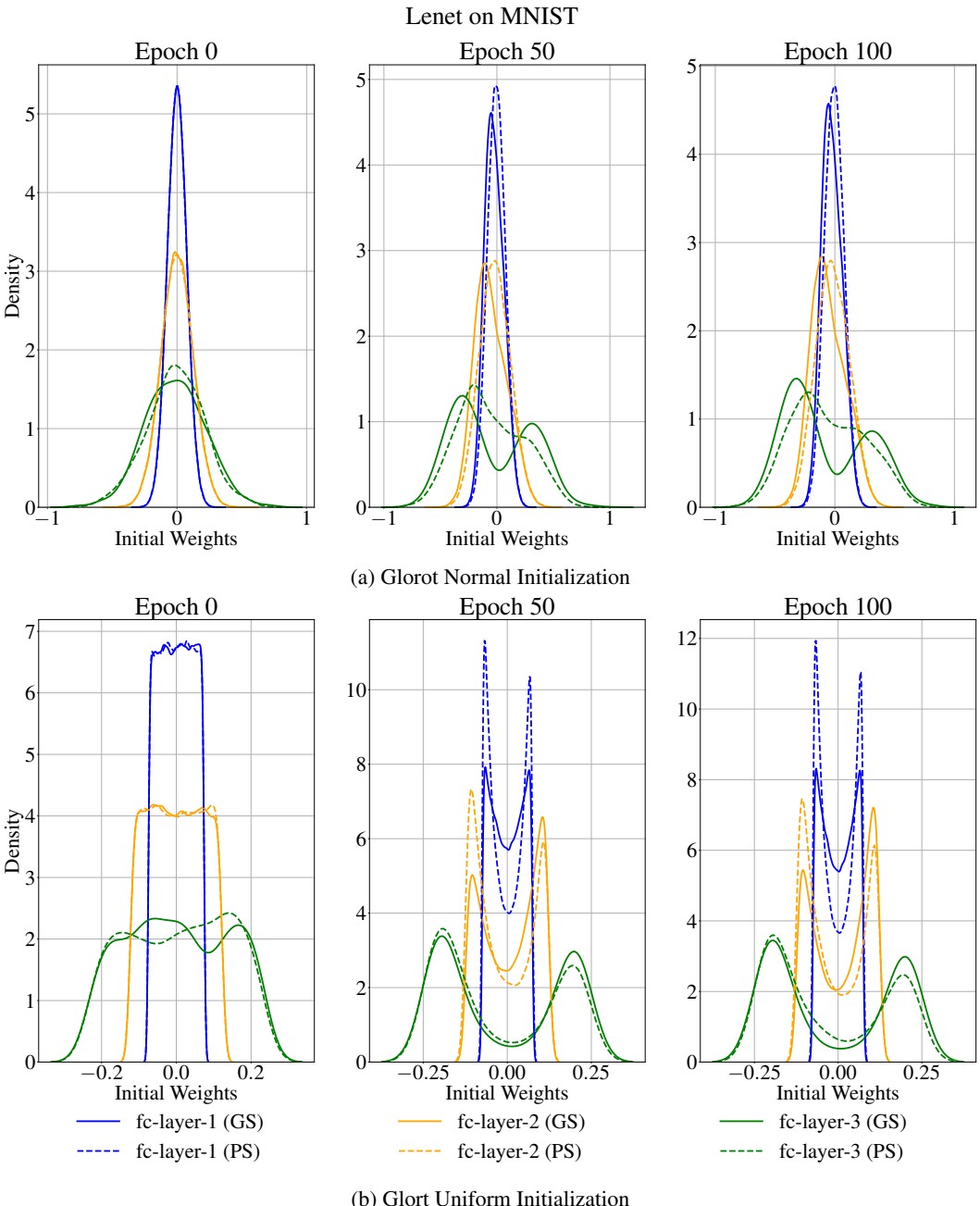

(a) Glorot Normal Initialization

(b) Glort Uniform Initialization

Figure 10: **Distribution of selected weights on MNIST.** As noted above, both sampling methods tend to choose larger magnitude weights as oppose to small values. This behavior is more evident when the values are sampled from a Glorot Uniform distribution (*bottom*) as opposed to a Glorot Normal distribution (*top*). However, layer 3 which has the fewest number of weights of any layer in this work continue to select large magnitude weights even when using a normal distribution.

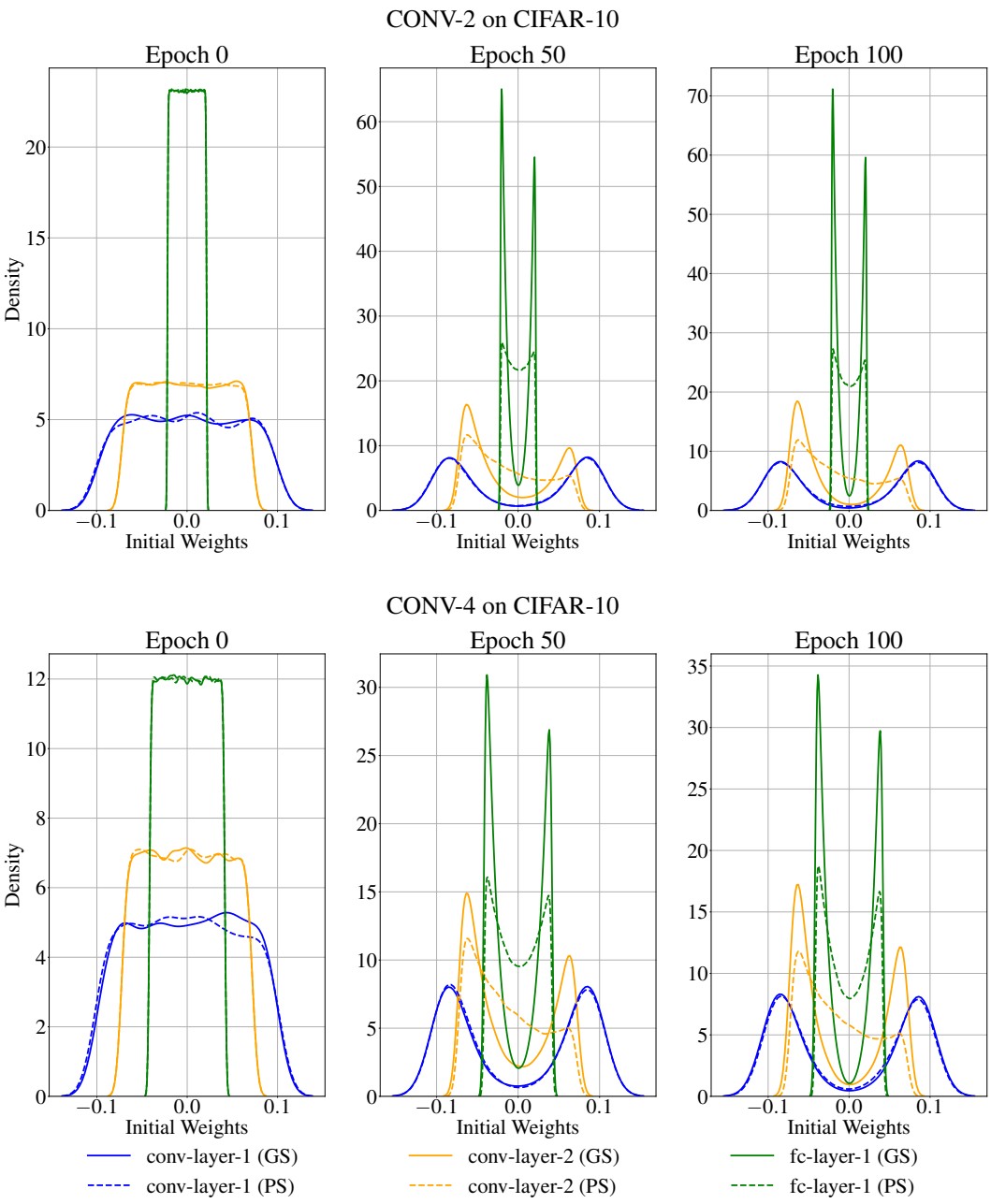

Figure 11: **Distribution of selected weights on CIFAR-10.** Similar to the plots shown in Figure 8, both CONV-2 and CONV-4 on CIFAR-10 tend to choose increasingly bigger weights in terms of magnitude as training progresses. Here, we show the distribution of the selected networks in the first two convolutional layers and the first fully-connected layer of the above networks but all the layers in all slot machines show a similar pattern.

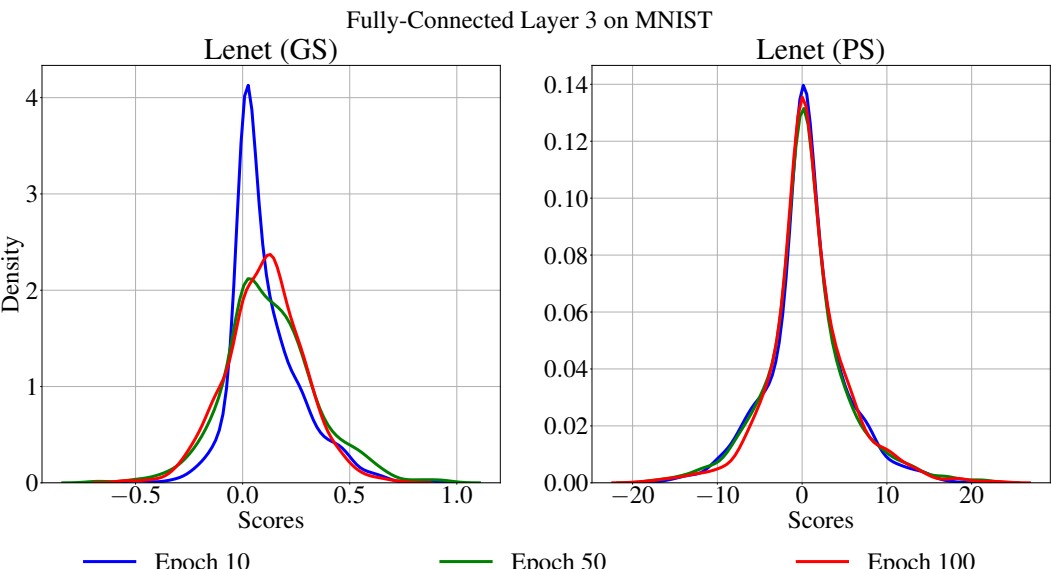

Figure 12: **Distribution of selected scores.** Different from the selected weights, the selected scores tend to be normally distributed for both GS and PS. We show only the scores for layer 3 of Lenet because it is the layer with the fewest number of weights. However, the other layers show a similar trend except that the selected scores in them have very narrow distributions which makes them uninteresting. Notice that although we sample the scores uniformly from the non-negative range $\mathbb{U}(0, 0.1 * \sigma_x)$ where $\sigma_x$ is the standard deviation of the Glorot Normal distribution, gradient descent is able to drive them into the negative region. The scores in PS slot machines move much farther away from the initialization compared to those in GS due to the large learning rates used in PS models.

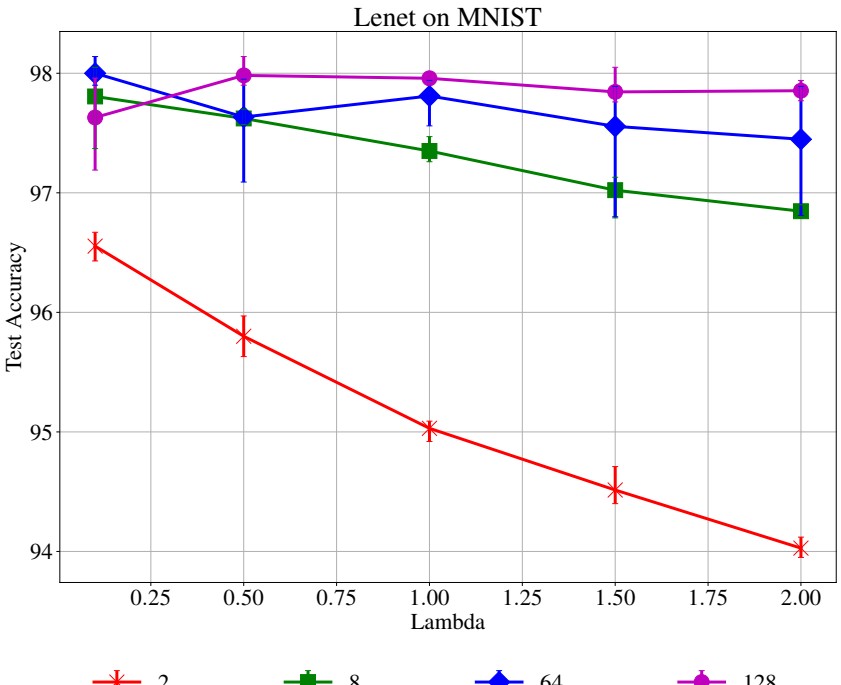

Figure 13: **Scores Initialization.** The models are sensitive to the range of the sampling distribution. As discussed in Section 4, the initial scores are sampled from the uniform distribution $\mathbb{U}(\gamma, \gamma + \lambda \sigma_x)$. The value of $\gamma$ does not affect performance, so we always set it to $0$. These plots are averages of $5$ different random initializations of Lenet on MNIST.

