# OpenReview forum: "Slot Machines: Discovering Winning Combinations of Random Weights in Neural Networks"
_ICLR.cc/2021/Conference — Reject_

### Official Review · AnonReviewer1 · 2020-10-27
**Simple idea, well executed, well presented, and with good results - but lacks a bit in novelty**

**Rating:** 7
**Confidence:** 4

**Review:**

**Update after authors' response**
I want to thank the authors for their response, and I am happy to see additional results on shared sets of weight values (which allows to easily relate the work to methods for training low-bit-width networks). To further increase impact and significance of the work it would be necessary to really flesh out the advantage of the proposed method over other, similar methods ("it is not too surprising that the method works, but why would I prefer it over other methods?"). Nonetheless, the paper presents novel empirical analysis that adds to the body of work on non-standard training of neural networks. To make a clear stance for the reviewer discussion I have therefore increased my score to 7, though I would rate the paper at the lower end of score-7 papers.
---

**Summary**
The paper proposes a novel scheme to obtain well performing neural networks without classical adaptation of weights. Instead, each connection can have one out of K randomly drawn values, and “training” consists of a backpropagation-based procedure to find which value out of the K possible values to select for each weight. The method can be interpreted as finding a high-performing sub-network within a larger network of random weights. However, in contrast to previous methods that literally implement the latter, the proposed method is computationally more efficient. Experiments are performed on a number of neural network architectures on MNIST and CIFAR-10.

---
**Contributions, Novelty, Impact**

1) Proposal of a novel scheme for finding well-performing networks without explicit training of weights. This is interesting and adds to a growing body of recent work on alternatives to classical training of neural nets (which is insightful for both, developing better training algorithms but also understanding the nature of neural network training). My concern is that the proposed method is conceptually very similar to previously known approaches (pruning a larger network which is also discussed in the paper, but also some methods for training low-bit-weight networks such as [1] and [2]). While the proposed method is an interesting alternative implementation, the advantages compared to the other approaches are fairly limited. Accordingly the potential impact of the work might be somewhat limited as well, I’m afraid.

2) A nice and extensive set of ablations and control experiments, as well as repetitions of experiments to establish statistical significance of the results. The paper, in particular the experimental part is well executed, and the ablations and controls allow for being optimistic about the generality of the findings, which has a positive influence on the potential impact of the work.

3) The paper shows that networks obtained with the proposed scheme can also act as a good initialization for further fine-tuning, leading to very well performing classifiers. This process is also analyzed in terms of overall computational budget (FLOPS) and in 2 of 3 cases shown compares favourably against standard neural network training. In terms of impact this is another nice result to add, but probably not strong enough to replace standard initialization anytime soon.


[1] Binarized Neural Networks: Training Deep Neural Networks with Weights and Activations Constrained to +1 or -1, Courbariaux et al. 2016

[2] XNOR-Net: ImageNet Classification Using Binary Convolutional Neural Networks, Rastegari et al. 2016

---
**Score and reasons for score**
I’m quite torn with this paper - on one hand the method works well, is thoroughly analysed and the paper is very well written and polished (in many respects I would even say this is an exemplary paper). On the other hand the paper suffers from only adding a quite simple variation on existing work. Particularly the work on training low-bit-width networks ([1] and [2] above, as well as later extensions to more than single-bit weights) is conceptually very similar - the forward pass uses constrained weight values but gradient information is accumulated in a real-valued variable. As the appendix notes, the main idea could also be implemented simply via pruning in a larger network (though less computationally efficient).

To further strengthen the paper it would be good if the paper could answer one (or both) of the following questions: (i) how do the results contribute to understanding weight initialization and the training process, what can the reader learn that wasn’t known already, (ii) what are the concrete advantages of the proposed method over previously proposed alternatives, what does it do better, what shortcomings does it address (is it faster, is training more stable, …)?

I would love to give this paper a very high score because of the great execution and presentation, but the lack of novel insights, or clear methodological advantages makes this hard. I am currently voting for a weak accept (because the paper is very well written and experiments are thorough, if the scoring was based on novelty alone I’m not sure that the paper would clear the bar for a top-tier conference). I am of course happy to reconsider my final verdict in light of the other reviews and authors’ response.

---
**Strengths**
1) Great presentation of the work, a very well written paper, and good main results.

2) Experiments are well executed - multiple repetitions, many controls and ablations that one wants to see to improve confidence in the generality of the findings.


---
**Weaknesses**
1) Little novelty. The proposed algorithm is a nice idea, but it’s not too surprising that it operates essentially on par with a previously proposed pruning method since the algorithm can even be conceptually recast as pruning in a larger network.

2) The writing puts a lot of emphasis and focus on the distinctive features of the method (understandably so given how close it is compared to other methods). But I think it’s also fine to not start off in an almost defensive fashion and simply state that this is another possibility of implementing the same idea but with the following advantages (and disadvantages).

---
**Correctness**
Reasoning in the paper is sound, experiments are well executed and many control-experiments and ablations are shown.

---
**Clarity**
The paper is very well written, results are nicely presented and related literature is discussed in a useful fashion.

---
**Improvements (that would make me raise my score) / major issues**

1) The main issue is a lack of novelty. Addressing either (i) how the paper adds new knowledge (in light of the current body of literature) or (ii) stating the precise advantages of the proposed method over alternatives would be crucial. I don’t see an obvious way for (i), but for (ii) a starting point could be to do more detailed comparison against other methods, in particular Ramanujan and see whether the proposed method compares favorably e.g. in terms of training stability of robustness w.r.t. hyper-parameters.

2) Another possibility to add novelty to the paper would be to focus on low-bit-width training, where instead of drawing K values for each weight separately, values are re-used per layer or even across the whole network (typically first and last layers need to be treated differently, i.e. they require more bit-width). Reliable and robust methods to obtain e.g. 2-, 4- or 8-bit networks is a timely and important topic, and the proposed method has potential to contribute to that body of work as well (though it is admittedly a bit of a deviation from the current story and main focus of the paper). I want to add this as a suggestion to the authors (it could work, but it could also severely reduce focus and clarity of the paper), not necessarily as an improvement that I’d expect to see.

---
**Minor comments**

A) The ICLR header seems to be missing in the PDF file.

B) In the probabilistic version of the method it might be worth experimenting with some “annealing” schedule where the randomness of the method is gradually reduced over training (e.g. via reducing a Gumbel Softmax temperature). Making the magnitude of the scores very large has essentially the same effect but is less controllable since it has to be indirectly influenced via the learning rate. I would expect convergence/test-time performance of the method to benefit from such a schedule and perhaps even help close the gap to “greedy selection”.

C) Is the supplementary material supposed to go into a separate document (did not double-check the submission instructions)?

D) A bit of a nitpick, the phrase that “weights are never updated” might suggest some miraculous phenomenon - I’d rather say there’s a set of weight values, from which one value is probabilistically selected. So if one considers the weight of a particular connection as a random variable (across different forward passes) then the expected value over that random variable changes (smoothly) as training progresses - resembling a standard weight update process quite closely.

---

> ### Author Response · Authors · 2020-11-21
> **Reply to Reviewer 1**
>
> We want to thank the reviewer for the many suggestions to improve our work.
>
> We agree with the reviewer that slot machines relate to low-bit networks. Both types of networks restrict the weights to assume values from a finite set. However, slot machines are different from low-bit networks in motivation and approach. Whereas the goal of low-bit networks is compression, the motivation behind slot machines is recovering effective weight configurations/initializations from small sets of random weights. Accordingly, the weights of low-bit networks are directly optimized whereas slot machines optimize only the scores associated with weights. Finally, slot machines use real-valued weights (because the objective is not compression) as opposed to the binary weights (-1, 0, +1) used in low-bit networks. We believe there are ample opportunities to push slot machines towards objectives such as compression. Accordingy, following the reviews we have added experiments described in Section 4.4 where all the connections use a common set of randomly generated weights, enabling a form of compression.

---

> > ### Comment · AnonReviewer1 · 2020-11-23
> > **Thanks for the response and the additional experiment**
> >
> > I want to thank the authors for their responses to all reviewers. I think the additional experiment with the same set of candidate-values shared across weights is interesting (and addresses comments by R4, R5, and me). I will update my original review after properly engaging with the other reviews and the corresponding responses.

---

### Official Review · AnonReviewer4 · 2020-10-28
**Weights obtained by the propoesd method are no longer completely random weights**

**Rating:** 4
**Confidence:** 2

**Review:**

##########################################################################

Summary:

This paper proposes a method to train a neural network by selecting a weight from a set of $K$ randomly generated weights for each edge in the network. Each edge has a different set of random weights. Quality score is assinged to each of $K$ random weights, which determines the weight used in the forward calculation. Instead of optimizing weights directly, the proposed method optimizes the quality scores with the straight-through gradient estimator. Experimental results show that the neural network trained by the proposed method achieves high accuracy compared to random initialization even when $K=2$.

##########################################################################

Reasons for score:

Overall, I vote for rejecting. The authors say in Sec. 3 that the goal is to construct non-sparse neural networks with completely random weights that achieve high accuracy. However, the model obtained by the proposed method is no longer a network with completely random weights, because the authors optimize quality scores instead of original weights. It is empirically shown that a neural network can achieve a high accuracy by properly selecting weights from a set of random weights prepared in advance. However, such a result is not so surprising from the viewpoint that the quality scores are optimized.
Also, this paper has few practical implications. I would like to see if the network can still achieve a high accuracy when every edge has a common set of $K$ random weights. If this is the case, the proposed method may lead to a network that is efficiently compressed.


##########################################################################

Minor concerns:

(p.1) a fixed a set of random weights -> a fixed set of random weights


--

Regarding the author responses, I have updated my rating.

---

> ### Author Response · Authors · 2020-11-21
> **Reply to Reviewer 4**
>
> - **Is weight selection a form of training?** As also  noted in our response to Reviewer 5 and Reviewer 3, we agree that weight selection is a form of training, a point we acknowledge when describing the analogy between our method and slot machines in the last paragraph of page 1. That said, we recognize that we probably placed too much emphasis on the fact that the weights themselves are never updated. We have revised the paper to reflect the feedback from the reviewers. The defining part of our work is that our method chooses a weight from a small set of weights for each connection, as opposed to the traditional continuous optimization of weights (SGD).

---

> > ### Comment · AnonReviewer4 · 2020-11-25
> > **Thank you for your updates**
> >
> > Thank you for your comments and updates.
> >
> > The modification in Sec 3 about the goal of the paper addresses one of my concerns. And additional experiments in Sec. 4.4 seems interesting.
> >
> > (Responses about "the preference for large weights is not surprising" may be not for me, but for other reviewers, because I did not comment on these points in my review.)

---

> > > ### Author Response · Authors · 2020-11-25
> > > **Thank you for your comment**
> > >
> > > Thank you for your comments and suggestions to improve the work.
> > >
> > > You are correct, the response about "preferences for large weights" was targeted at a question Reviewer 3 raised. We have updated our responses accordingly.
> > >
> > > Thank you.

---

### Official Review · AnonReviewer3 · 2020-10-28
**Instead of updating weights, we can train a network by picking from a set of (random) weights.**

**Rating:** 5
**Confidence:** 3

**Review:**

### Summary

The paper investigates a type of neural network in which one of K possible fixed weights is chosen in each neuronal connection. The weights themselves are fixed and random, but the scores that determine which of the weights is chosen are updated through back-propagation using a straight-through estimator. The accuracy and behavior of such networks is studied for small networks on small datasets (MNIST, CIFAR).

### Quality, clarity, originality and significance

The paper is well-written and easy to follow.

The main idea seems interesting at first sight and it is well-motivated, but after some consideration of related effects in neural networks, the results do not seem very surprising (see below). I think a deeper exploration of the connection to other phenomena would be necessary to make this paper relevant to the conference, e.g. to weight quantization (to few bits per weight) or to (variational) dropout. The paper seems to not go beyond ad-hoc conclusions of the form that these (peculiar) networks "perform competitively on challenging datasets" (which seems to be a bit of an overstatement to me). The authors also claim that the trained networks might be useful for initialization, but to really make this point strongly, a comparison to other practical methods of data-driven initialization on larger datasets with larger architectures might be needed to convince the reader.

Why do the results not seem surprising to me: It is possible that I misunderstood the algorithm (which we could clarify in the rebuttal period, of course), but in my understanding of the described approach, the straight-through estimation of the scores will lead to preferring the selection of larger (or smaller) weights where standard gradient descent training would lead to larger (or smaller) weights. This is consistent with the distribution of the selected weights as shown in several figures, with the observation that uniform initialization works better than Normal initialization, and with the observation that "both GS and PS tend to prefer weights having large magnitudes as learning progresses". It would also account for the observation of the similarity in error rates when the network has a sufficient number of weights to choose from.

* Pros: interesting idea, the paper is well-written
* Cons: of limited interest to the conference audience I believe; not clear if there is practical relevance or potential to improve scientific understanding

### Detailed/minor comments
* The main bullet points at the end of the introduction were not fully substantiated in the paper in my opinion: (1) I was not fully convinced of "a performance equivalence between random initialization and training", because the slot machines are effectively *trained*. (2)"demonstrates that current networks can model challenging non-linear mappings extremely well even with random weights" is not 100% clear because the weights are a *choice* among a set of *initially* random weights, but that choice is a result of training. (3)"connects to recent observations". This seems to happen mostly in the two sentences before 4.3 and there connection *and* statement are not entirely convincing to me "We find a random 6 layer network that performs as well as a 6 layer trained network without any form of pruning." Here it seems to me that the slot machine after training is in fact not random, because it was trained, and that training exploited weight-correlations that potentially span multiple layers. The argument could be extended to regular training in the sense that regular training just picks out the random weights among all the random floating point numbers. (This is a bit exaggerated, but I think it shows why the term "random 6 layer network" may be an overstatement after training.)
* Why is K chosen from the set {2, 8, 64, 128} - it seems that some natural values in this sequence are missing or that a more log-uniform spread would be more "natural". Maybe there is a specific reason that is not obvious, then it could be mentioned.

### Update after author replies and discussion
I have updated the review score after reading the authors' reply and revision of the paper.

---

> ### Author Response · Authors · 2020-11-21
> **Reply to Reviewer 3**
>
> - **Is weight selection a form of training?** We agree with the reviewer that we optimize the selected network through the scores, a point we probably did not emphasize enough in the paper. We focused on the fact that the weights are never updated after the random initialization. We did not in anyway intend to suggest or create the impression that the networks are not trained since the selection process is a form of training albeit a different procedure compared to traditional SGD. We have revised the paper accordingly.
> - **Comparison to network quantization?** Following the reviewer's feedback, we have included experiments, described in Section 4.4,  (1) where all the connections in a layer share a common set of $K$ weights and (2) where all the connections in the network share a common set of $K$ weights. These networks, like the conventional slot machines also obtain high accuracy. We refer to Section 4.4. for further insights.
> - **The preference for large weights is not surprising.** Our experiments suggests that large magnitude weights are preferred over time. However, there is no procedural reason why the method would select according to magnitude. We also do not have any evidence suggesting that slot machines choose networks similar to those that standard backprogation would yield. As described in Section 3.1, slot machines effectively operate by selecting high performing networks from many randomly initialized networks by means of loss optimization.

---

> > ### Comment · AnonReviewer3 · 2020-11-24
> > **Thank you for your reply**
> >
> > Thank you for replying to the review and for updating the paper, including the new experiments in Section 4.4, which address one point that has been raised by several reviewers. We will take all this into consideration during the discussion period.

---

### Official Review · AnonReviewer5 · 2020-11-06
**Results interesting but not insightful. Motivation unclear. Some issue with scientific method.**

**Rating:** 6
**Confidence:** 4

**Review:**

Summary: The work extends an existing algorithm to train a convolutional neural network by selecting a subset of fixed random weights by (1) using multiple random values per weight, (2) using a sampling procedure to select the particular random value for each weight. If these networks are finetuned after random-value selection is performed, they perform close to networks purely trained with SGD on CIFAR-10 and MNIST.

Strong points:
- Interesting results that extend results on the lottery ticket hypothesis and random weight learning.

Weak points:
- The paper builds heavily on an existing algorithm, but it does not cite it in the method section.
- The paper makes many claims and fails to provide evidence for these claims.
- The work is not clearly motivated. It is unclear why this problem is interesting or the results insightful.

Recommendation (short):
While the results are interesting, the paper is poorly motivated and does not conform to standards of scientific work. I recommend rejecting this work.

Recommendation (long):
The method section appears to be independent work, but it is the same as Ramanujan et al., 2019 extended with multinomial sampling. While Ramanujan et al., 2019 are cited, it is not cited in this section. Furthermore, the paper claims that random initialized networks and trained networks perform the same, but it does not lay out evidence or an argument for this. Otherwise, it is unclear why this method is interesting. I recommend rejecting this work.

Comments for authors:

I think these are some good initial results that you have, and you can work with that, but right now, the paper has many flaws that need to be ironed out before you make another attempt. I do not think you can get this work accepted in this round and instead should try to learn as much as possible from the discussion for a resubmission.

I think fixing the claim, references, and so forth will be easy. The hard question is, why is your work interesting. Using multiple random values per weight can be seen as some form of quantization
of weights. Why is your way of doing something similar to quantization more interesting than other forms of quantization?

Please also consider if it is true that selecting fixed random weights is training or not. Clearly, you are optimizing weights. It does not matter if you do it with SGD, an evolutionary strategy, or your algorithm; in the end, you optimize weights to have particular values. I would say that it should be considered training. But if your method is considered just a different optimization process compared to SGD, why is it interesting? Finding subnetworks, like done in Ramanujan et al., 2019, is interesting because you have smaller trained networks, but you do not have subnetworks.

It could be interesting if you do (1) a thorough analysis that yields some insights and make this an analysis paper, or (2) try to get better performance by doing both optimizations of weights and selection of weights (but this is very similar to Wortsman et al. 2019).

Some minor things:
- Equation 3 has an additional W (h(*) already contains the W)
- Figure 6 has an annotation error; I believe the upper line is supposed to be the PS method


Ramanujan et al., 2019: What’s Hidden in a Randomly Weighted Neural Network?, https://arxiv.org/abs/1911.13299
Wortsman et al., 2019: Discovering Neural Wirings, https://arxiv.org/abs/1906.00586

---

> ### Author Response · Authors · 2020-11-21
> **Reply to Reviewer 5**
>
> - **Relation to Ramanujan et al. (2019)?** We have been heavily inspired by the impactful work of Ramanujan et al. (2019), as evidenced by the fact that it is the most cited paper in our submission. At the same time, our approach differs in important ways.  Ramanujan et al. (2019) use $1$ weight per connection whereas we use $K$ weights per connection.  Additionally, the method in Ramanujan et al. (2019) operates by pruning connections. Our algorithm does not prune any connection—it simply selects $1$ of $K$  weights assigned to each connection. We believe we properly acknowledged the influence of Ramanujan et al. (2019) on our work throughout the paper. We note that in the technical section we cite the straight-through gradient estimator by Bengio et al. (2013) (instead of  Ramanujan et al. (2019)) since this is the actual method we employ in our optimization.  Bengio et al. (2013) was also used and cited in Ramanujan et al. (2019).
> - **Is weight selection a form of training?**  We agree with the reviewer that, although the weights themselves are never changed, our method does perform an optimization by updating the scores. We have revised the paper to properly acknowledge that this optimization clearly constitutes a form of training, albeit different from that traditionally done by updating weights.
> - **Why is this work interesting?** It provides insights about the strong performance that networks can achieve by means of selection from fixed random weights. We believe that while pruning has recently emerged as as a powerful optimization mechanism, selection from fixed sets of weights is still a relatively unexplored strategy. We note that our approach differs from network quantization methods, in terms of both goals as well as methods. While algorithms for low-bit or quantized networks aim at compressing the size of the model, the objective of our approach is to uncover strong configurations or initializations from small sets of weights. Furthermore, while most network quantization methods operate by updating the weights, our optimization is carried out on scores associated to fixed weights. In this revision, we have added a paragraph discussing network quantization methods in the related work section. We have also added a new subsection (4.4) presenting results for slot machines optimized under the constraint of shared weights, which is effectively used in network quantization methods.
>
>    $\quad$
>
>    Finally, we hope that our empirical study showing the existence of random weight configurations with impressive performance will spur further research on network initialization. We note, as an example, that the lottery ticket hypothesis (Frankle & Carbin, 2018) has been highly influential in the area of training subnetworks. However, the hypothesis by itself does not enable any immediate practical use due to the high computational cost imposed by having to train the full network. Furthermore, it hinges on a pruning method that has been known for years. Despite all this, it has been a source of inspiration for a flurry of exciting new work. We highlight this example not to suggest any equivalence of potential impact between the lottery ticket hypothesis and our work but rather to underscore that immediate practical application should not be the sole determinant of relevance.
>
> **References**
>
> Yoshua  Bengio,  Nicholas  L ́eonard,  and  Aaron  Courville.    Estimating  or  propagating  gradientsthrough stochastic neurons for conditional computation, 2013.
>
> Jonathan Frankle and Michael Carbin. The lottery ticket hypothesis: Finding sparse, trainable neuralnetworks, 2018.
>
> Vivek Ramanujan, Mitchell Wortsman, Aniruddha Kembhavi, Ali Farhadi, and Mohammad Raste-gari. What’s hidden in a randomly weighted neural network?, 2019.

---

### Decision · Program_Chairs · 2021-01-07
**Final Decision**

**Decision:**

Reject

**Comment:**

The idea behind this paper is to develop a training algorithm that chooses among a fixed set of weights for each true weight in a neural network.  The results are reasonable -- though difficult to quantify as either good or surprising -- performance from the algorithm. A perhaps interesting point is that additional fine-tuning from these found networks can, in some cases, best the accuracy of the original network.

The pros of this paper are that it is a neat original idea. With the exception of the limited scale of the benchmarks (i.e., the selected architectures), the paper is largely well-executed.  The primary shortcoming of the paper, as discussed by the reviewers, is the lack of clarity in its implications. Specifically, it is difficult to position the result as contributing to a practical aim or leading to additional future work.

Based on the reviews and discussion, my recommendation is Reject. In particular, this paper would be significantly improved by bringing in a strong motivational context and, therefore, additional comparisons.

For example, the context for the work of Ramanujan et al. (2019) is that, perhaps, it is possible to find subnetworks of large initialized networks that will permit more efficient training. In Appendix A, this paper proposes that the technique here could be cast as pruning within a much larger network. Following results from Zhu and Gupta [1] and also Ramanujan et al. (2019), finding a sparse network within a larger network can produce a more accurate network than training a network of equivalent size to the sparse. Therefore, these results could, potentially, be cast and as a more efficient way to perform the techniques of Ramanujan et al. (2019).

Alternatively, the results that demonstrate that fine-tuning the identified networks improves performance over the standard network could be more robustly evaluated and perhaps cast as either an alternative training technique or leveraged as a technique like warm starting [2].

This is a very interesting and promising direction. It appears that the paper just needs a bit more distillation.

[1] To prune, or not to prune: Exploring the efficacy of pruning for model compression. Michael Zhu and Suyog Gupta. In International Conference on Learning Representations Workshop Track, 2018.

[2] On Warm-Starting Neural Network Training. Jordan T. Ash, Ryan P. Adams. NeurIPS 2020